



**Assessing the degree of plug flow in oxidation flow reactors (OFRs): a study on a Potential**
**Aerosol Mass (PAM) reactor**
Dhruv Mitroo[1,a,*], Yujian Sun[1,*], Daniel P. Combest[2], Purushottam Kumar[3], and Brent J.
Williams[1]
[1]Department of Energy, Environmental & Chemical Engineering, Washington University in St.
Louis, St. Louis, MO, USA
[2]ENGYS, Ltd.
[3]Discipline of Chemical Engineering, Indian Institute of Technology Gandhinagar, Palaj, Gujarat
382355, India
[a]Now at the Department of Atmospheric Sciences, Rosenstiel School of Marine and Atmospheric
Sciences, University of Miami, Miami, FL, USA
[*]equally contributing authors
Correspondence to: brentw@wustl.edu
Brent J. Williams, Ph.D.
Raymond R. Tucker I-CARES Career Development Associate Professor
Department of Energy, Environmental & Chemical Engineering
Washington University in St. Louis





**Abstract**
Oxidation flow reactors (OFRs) have been developed to achieve high degrees of oxidant exposures
over relatively short space times (defined as the ratio of reactor volume to the volumetric flowrate).
While, due to their increased use, attention has been paid to their ability to replicate realistic
tropospheric reactions by modeling the chemistry inside the reactor, there is a desire to customize
flow patterns. This work demonstrates the importance of decoupling tracer signal of the reactor
from that of the tubing when experimentally obtaining these flow patterns. We modeled the
residence time distributions (RTDs) inside the Washington University Potential Aerosol Mass
(WU-PAM) reactor, an OFR, for a simple set of configurations by applying the tank-in-series (TIS)
model, a one parameter model, to a deconvolution algorithm. The value of the parameter, $N$, is
close to unity for every case except one having the highest space time. Combined, the results
suggest that volumetric flowrate affects mixing patterns more than use of our internals. We
selected results from the simplest case, at 78s space time with one inlet and one outlet, absent of
baffles and spargers, and compared the experimental F-Curve to that of a computational fluid
dynamics (CFD) simulation. The F-Curves, which represents the cumulative time spent in the
reactor by flowing material, match reasonably well. We value that the use of a small aspect ratio
reactor such as the WU-PAM reduces wall interactions, and suggest applying the methodology of
tracer testing described in this work to investigate RTDs in OFRs and modify inlets, outlets, and
use of internals prior to applications (e.g., field deployment vs. laboratory study).

**1 Introduction**





Tubular reactors were first introduced to the field of atmospheric science by means of small flow
cell reactors developed to study the kinetics of stratospheric reactions (Brune et al., 1983; Howard,
1979; Keyser, 1980; Lamb et al., 1983). Accurate kinetic measurements were possible due to the
high pipe aspect ratios, which encouraged a high degree of plug flow behavior (Keyser, 1984).
The design of these miniature tubular reactors, with volumes on the order of a few cm$^3$, was
different from that of significantly larger, batch-type or semi-continuous type well mixed reactors,
with volumes on the order of several m$^3$, built to understand aerosol formation in the troposphere
(Crump et al., 1982; Crump and Seinfeld, 1980; Leone et al., 1985). To study aerosol formation
and growth chemistry, the dynamics of atmospheric circulation and transport needed to be
excluded. It was therefore convenient to mimic the troposphere by treating it as an enormous, well
mixed reactor, which led to the development of larger well mixed reactors. The discovery of
secondary processes preceding aerosol formation led to significant emphasis on the study of
secondary organic aerosol (SOA) formation (Haagen-Smit, 1952, 1963, 1970; Went, 1960). The
approach of using large, well mixed batch-style environmental chambers eventually helped
elucidate chemical mechanisms for model compounds (Claeys, 2004; Kamens et al., 1982; Kroll
et al., 2006; Nozière et al., 1999; Paulson et al., 1990; Pereira et al., 2015; Volkamer et al., 2001),
and, with improved instrumentation (Canagaratna et al., 2007; Crounse et al., 2006; de Gouw and
Warneke, 2007; Hansel et al., 1995; Jayne et al., 2000; Williams et al., 2006; Zhao et al., 2013),
the community gained a better understanding of SOA formation. Unfortunately, low levels of
conversion and high wall losses seen in these large reactors did not allow simulated exposures that
exceeded a day at most, which is just a short glimpse into the average two week lifespan of an
atmospheric aerosol (Seinfeld and Pandis, 2006). Due to such limitations, oxidation flow reactors
(OFRs) with short spacetimes (ratio of reactor volume to the volumetric flowrate) are being



developed (Cazorla and Brune, 2010; Ezell et al., 2010; George et al., 2007; Huang et al., 2016;
Kang et al., 2007).

OFRs can be viewed as tubular reactors due to their pipe aspect. They have been widely used for
over a decade to study heterogeneous reactions on organic aerosol surfaces involving gas-phase
oxidants such as hydroxyl radicals and ozone (George et al., 2007; George and Abbatt, 2010;
Katrib et al., 2005; Kessler et al., 2010, 2012; Knopf et al., 2005; Kroll et al., 2012; Smith et al.,
2009). These reactors are able to generate very high concentrations of hydroxyl (OH) radicals, tens
to thousands times higher than typical tropospheric levels, which accelerates the rate of gas-phase
oxidation reactions. Within spacetimes of a few minutes, it is possible to achieve integrated oxidant
exposures equivalent to multiple days or weeks of atmospheric oxidation. It is important to
distinguish OFRs from modern day conventional flow tube reactors, which stem from designs of
old flow tube reactors (e.g., Keyser 1984) but employed in the study of gas uptake kinetics on
aerosol surfaces rather than homogeneous gas-phase reactions, as described in the previous
paragraph. Beyond the original application of heterogeneous oxidation studies, Kang et al.
introduced the potential aerosol mass (PAM) OFR which, alongside newer OFR designs, was
intended specifically for studies of SOA physicochemical properties (Kang et al., 2007, 2011;
Keller and Burtscher, 2012; Lambe et al., 2011b, 2012, 2013; Massoli et al., 2010; Ortega et al.,
2013; Slowik et al., 2012). This application therefore altered the study of SOA formation,
previously dominated by the traditional large, well mixed reactors (Kroll and Seinfeld, 2008;
Rudich et al., 2007; Turpin et al., 2000), by allowing to generate laboratory data beyond first
simulated day of exposure. Because the mechanism of exposure between traditional chambers
OFRs was different, validating the OFR concept began by replicating data obtained from



traditional chambers (Chhabra et al., 2015; Lambe et al., 2015; Liu et al., 2015), and to assess
whether the chemistry was realistic (Li et al., 2015; McNeill et al., 2008; Peng et al., 2015;
Renbaum and Smith, 2011). Consequently, much modeling work has focused on pure chemical
reactions and comparison of SOA yields between the two (Bruns et al., 2015; Lambe et al., 2015;
Li et al., 2015; Ortega et al., 2015; Peng et al., 2015). However, essentially no modeling work has
been done on understanding hydrodynamics or flow fields inside OFRs so that the flow patterns
can be improved. In a study from Li et al., it appears that residence time distributions (RTDs) that
deviate significantly from plug flow in the PAM result only in a ~10% error of reported values
such as OH exposure (Li et al., 2015), which is conducive to OFRs being viewed as tubular
reactors. It is unknown if the error may trend with external OH reactivity ($OHR_{ext}$) and become
more significant for slow reacting compounds, although efforts by the Jimenez Group at the
University of Colorado at Boulder are underway.

In both single and multiphase reactors, contact patterns and the degree of mixing determine reactor
performance, e.g., selectivity and yield (Bourne, 2003; Deckwer, 1976; Levenspiel, 1999). This
implies that upon desired contacting, chemical pathways that would be otherwise suppressed can
become more competitive. For example, if during a mixed OH / ozonolysis heterogeneous
reaction, a fresh biomass burning aerosol is introduced in the centerline port of an OFR and ozone
is introduced along a side port, most of the aerosol may travel ballistically through the chamber
having limited contact with ozone or OH, and chemical reaction is less competitive with photolysis
/ photobleaching reactions of the aerosol. RTDs describe the probability of a fluid element's age
inside the reactor: one can think of those as the probability distribution function (PDF) of a fluid
element in the reactor (Fogler, 2006; Levenspiel, 1999). Tools are available to diagnose or predict



flow behavior. These tools fall in two categories: tracer tests (diagnostics) and computational fluid
dynamics (CFD) simulations (predictions).

We present a technique to assess the degree of plug flow in an OFR, that can be in principle
extended to any vessel. The rigor of the technique is tested by varying use of internals and flowrate
and observing the resulting RTD curves in the Washington University PAM (WU-PAM) reactor.
We begin by introducing an experimental method for obtaining the reactor RTD, which can be
applied to any other OFR, via inert tracer injections. From raw data, we explain how to obtain
PDFs. We chose to run CFD on the simplest design (a base case configuration) of the WU-PAM
reactor to gain hydrodynamics information. Finally, we compare results from tracer tests and CFD
for the base case. We compare this approach to that of previous studies by Lambe et al. (Lambe et
al., 2011a), Huang et al. (Huang et al., 2016), and Simonen et al. (Simonen et al., 2016),to the best
of our knowledge, the only other studies on RTDs in OFRs. We do not provide predictive
configurations for the PAM reactor because there are many avenues different groups can take
depending on their focus, and this study is central to the current design.

**2 Methods**

The WU-PAM reactor is an iridite-treated aluminum cylinder, 18 inches in length and 8 inches in
inner diameter, giving it a total volume of 13 L. It has two 12 inch mercury lamps with peak
wavelengths at 185 nm and 254 nm (BHK Inc. Analamp Model No. 82-9304-03) housed in Teflon
sheaths, directly opposite each other, along the axial direction. Annular flow of $N_2$ (Airgas)
through the sheaths prevents direct contact with the lamps and purges any outgas products when





the lamps are turned on. The mercury lamps are left in place with their housing to mimic simple
OFR internals; they have not been turned on during this study. Details of their mode of operations
for oxidant formation can be found elsewhere (Li et al., 2015; Peng et al., 2015, 2016). OFRs like
the WU-PAM have removable internals, face plates, and peripheral inlets and outlets that allow a
wide variety of configurations. For example, Ortega et al. removed the inlet plate of their PAM
reactor during a deployment in the Fire Lab at Missoula Experiment (FLAME-3) while keeping
the inlet baffle to reduce particle loss, and in doing so observed a reduction in jetting of centerline
flow (Ortega et al., 2013). In a different study, Lambe et al. ran experiments keeping the inlet plate
on the PAM coupled with a sparger (a cap with large holes in the side in fixed onto the inlet, so
that the flow does not jet into the chamber), because laboratory experiments required a closed
system (Lambe et al., 2011a).

In this work, we chose four configurations: I (one inlet, one outlet, two lamp housings as internals),
II (one inlet, one outlet, two lamp housings with sparger and baffles as internals), III (multiple
inlets, multiple outlets, two lamp housings as internals), and IV (multiple inlets, one outlet, two
lamp housings with sparger and baffles as internals). Configuration I at 78s spacetime was subject
to a CFD simulation as a simple scenario where the simulation could capture hydrodynamics
accurately.

**2.1 Tracer studies**

The laboratory setup to determine RTDs experimentally is shown in Fig. 1. $N_2$ (Airgas) was the
carrier fluid and $SO_2$ (Air Liquide) was the inert tracer. Both flow rates were controlled by mass



flow controllers (MFCs) (Pneucleus Technologies, LLC). All experiments began by allowing one
hour to achieve a steady state of the carrier gas' flow profile inside the reactor, after which $SO_2$
was introduced in a single step-up manner. A tracer flowrate of 100 $cm^3$ $min^{-1}$ allowed good
detection in the measurement and minimized perturbation of the flow field. Analogously, the flow
of the carrier fluid was stepped down to maintain a constant desired total volumetric flowrate. $SO_2$
mixing ratios were determined by a Trace level-Enhanced $SO_2$ Analyzer (Thermo Scientific Model
43$i$, Thermo Scientific) via pulsed fluorescence, and the instrument was set to an averaging time
of 10s. This setting was the highest frequency over which the instrument could average the signal.
Obtaining high frequency data simplifies data analysis by avoiding the need for interpolation
techniques, as discussed in Sect. S1.

We expected that the tracer would experience an associated spacetime and RTD in places other
than the reactor, between the exit of the flow controller and the $SO_2$ detection chamber in the gas
analyzer. We therefore ran two experiments for every WU-PAM reactor configuration. The first
incorporated both the reactor and the inlet and outlet plumbing, and the second bypassed the
reactor. From these two signals we could extract the actual reactor RTD as described in Sect. 3.2.
Both experiments were operated by allowing the formation of fully developed flow before
injecting the tracer stepwise, as mentioned previously. Appendix A describes in detail how we
obtained a PDF and a cumulative distribution function (CDF) from raw data.

The WU-PAM reactor has peripheral inlets and outlets to optionally create allow a ring (annular)
flow around the centerline. Ideally, a uniformly distributed flow around the centerline helps
stabilize the flow, avoids recirculation, and reduces wall losses. To create ring flow, we formed a





three-eighth inch Teflon tube into a circle, and drilled six one-sixteenth inch diameter holes evenly
spaced along the side of the tube facing in the direction of flow. A similar Teflon tube circle was
created for the outflow. The ring flow setup required additional plumbing internals (Fig. 1b).
Tracer tests were accomplished for configuration I at three different spacetimes ( of 52s, 78s, and
152s), for three different configurations ( I, II, and III) at a 78s spacetime, and an arbitrary special
case for configuration IV at 411s spacetime (configuration and spacetime not commonly used).

**2.2 Simulations**

While tracer studies are a powerful diagnostic tool and result, if done correctly, in accurate RTDs,
they cannot capture the full hydrodynamics details, or the state of mixing in the reactor (i.e., the
exchange of mass between the fluid elements). Both hydrodynamics and mixing can significantly
influence the reactor performance (Fogler, 2006; Villermaux, 1986). For configuration I at 78s
spacetime, we ran a CFD simulation to visualize the hydrodynamics inside the WU-PAM. This
comparative analysis seeks to provide validation prior to using the CFD platform as a predictive
tool for mixing patterns in OFRs with more complex geometry or internals.

As a solver, we used OpenFOAM, an open source CFD toolbox available at www.openfoam.com
or www.openfoam.org. The reactor geometries were constructed on FreeCAD, an open source
computer aided design (CAD) software available at www.freecadweb.org, and Onshape, available
at www.onshape.com, prior to being exported into OpenFOAM. To discretize the volume elements
in the geometry, a mesh was created using the snappyHexMesh tool in OpenFOAM either directly
or in the HELYX-OS GUI. By generating mainly hexahedral meshes, this tool can mesh objects



of irregular shape. Then, additional layers of different geometry are added to the surface to improve
the mesh quality. A figure and details of the mesh can be found in Figure S1 and Table S1,
respectively. The hydrodynamics were calculated using simpleFoam, a steady-state solver for
single phase incompressible laminar or turbulent flow. We used first order schemes, and specified
the boundary conditions in each simulation case. The outlets had zero gradient for velocity and
fixed values for pressure, while the walls had fixed value for velocity and zero gradient for
pressure. After the flow field is obtained, a tracer experiment is simulated by scalarTransportFoam
for one of the simulations, which solves the transient convection-diffusion transport equation of a
passive scalar (dimensionless tracer concentration in this case). The initial condition is zero
concentration, and the boundary condition at the inlet is that the dimensionless tracer concentration
is equal to 1. After the simulation, the exit concentration is mixing-cup averaged to output an F-
Curve. We added a modification to the existing solver to account for turbulent diffusivity, which
had a non-negligible effect on mixing in the WU-PAM reactor, particularly at the entrance jet for
high flowrates. We found that the turbulent diffusivity was on the same order of magnitude as the
molecular diffusivity within the jet region near the inlet, suggesting turbulence in the jet was
significant. It is worthwhile to note that the inlet sparger and baffles (i.e., internals present in
configuration II and IV) left out of the simulation could significantly affect this outcome, however
those simulations required significant computer time to resolve mesh sizing.

**3 Results**

**3. 1 The RTD function, E(t), and the cumulative RTD function, F(t)**





Tracer tests give us fast qualitative information about the reactor, but mathematical manipulation
(e.g., normalizing the data and scaling the axes) of the data provide quantitative information and
offers a basis for comparing reactor behaviors on a universal scale. The main mathematical
descriptors of a fluid element residing in a chamber are its PDF and its CDF. For a chemical
reactor, the PDF is more commonly referred to as the RTD function, $E(t)$, in the dimensional
domain, or $E(\theta)$ in the dimensionless domain (referred to as E-Curves). Similarly, the CDF is
called the cumulative RTD function, $F(t)$, in the dimensional domain, or $F(\theta)$ in the
dimensionless domain (referred to as F-Curves) (Danckwerts, 1953; MacMullin and Weber Jr.,
1935). The relations between E-Curves and F-Curves are derived for the reader in this Appendix
A, but are well established and available on the internet and in classical textbooks (Fogler, 2006;
Levenspiel, 1999, 2002).

Figure 2 gives an example of how mathematical processing of the data looks. The shape of the
curve does not change, but the axes do. Section S1 explains how we obtained a pulse response
equivalent of concentration data from stepwise addition of the tracer.

In the WU-PAM, advective flow should be the main form of transport (we do not consider
convective effects due to thermal gradients from lamp activity in this work). Modeling real reactors
can be challenging, but approximations are possible using ideal reactor concepts (Levenspiel,
2002). The two most common examples of ideal reactors are the plug flow reactor (PFR), where
the flow is perfectly plugged or piston-like, and the continuously stirred tank reactor (CSTR),
where the flow is perfectly mixed. Mathematically, their E-Curves are represented by Equations

253 1-4:






$$E_{PFR}(t) = \delta(t - \bar{t}) \tag{1}$$

$$E_{PFR}(\theta) = \delta(\theta - 1) \tag{2}$$

$$E_{CSTR}(t) = \frac{1}{\bar{t}} e^{-\frac{t}{\bar{t}}} \tag{3}$$

$$E_{CSTR}(\theta) = e^{-\theta}. \tag{4}$$


Examples of how RTDs look like based on compartmental modeling using both ideal reactors are
available in chemical engineering textbooks (Fogler, 2006; Levenspiel, 1999) and, although not
discussed here, a variety of phenomenological models can be applied to describe or compare
OFRs. It is then open to interpretation whether the combination of ideal reactors chosen for an E-
Curve (e.g., a PFR and CSTR in series, or two CSTRs in parallel) describes the hydrodynamics of
the reactor as well. The RTD of an OFR should be obtained experimentally, if possible, before
deciding what model to use to describe it. Development of a phenomenological model to describe
the WU-PAM RTD is beyond the scope of this study, whose aim is to develop a robust
methodology to assess degree of plug flow in any OFR, however is an avenue that should be
pursued in the future. Given our current setup at Washington University, the true reactor RTD is
impossible to measure accurately by a single tracer injection. The tubing length, pressure drop
inside the filter holder upstream of the $SO_2$ detector, and location of the $SO_2$ detector have not
been minimized, thus we expect that collectively they could perturb our measurements
significantly. We choose not to simply subtract the theoretical space time of the tubing, because
non-ideal tracer injection or detection are most likely not represented by a Dirac function of a
perfect impulse (or derived from a perfect stepwise injection, represented by the Heaviside



function). Therefore we need to deconvolute the RTD signal due to the reactor from the signal due
to additional plumbing.

**3.2 Tank-in-Series model for indirect deconvolution**

Levenspiel describes the convolution integral (Levenspiel, 1999) in his textbook "Chemical
Reaction Engineering", which has been adapted to solve previous problems of decoupling RTD
signals (Hamed, 2012; Han, 2007; Mills and Duduković, 1988; Simonen et al., 2016; Sun, 2010).
This integral focuses on packets of the tracer that enter $t'$ seconds before $t$, that is $(t - t')$, and
stay $t'$ seconds in the reactor:

$$C_{out}(t) = \int_0^t C_{in}(t') \cdot E(t - t')dt', \qquad (5)$$

or

$$C_{out}(t) = C_{in} * E \qquad (6)$$

where $E$ is the true E-Curve of the reactor, and $C_{in}$ and $C_{out}$ are the time-dependent concentration
profiles of the measured tracer at the injection port and outlet port respectively. This equation is
based on assumptions of mass conservation (i.e., no wall loss inside the reactor) and memory loss
(i.e., the fluid elements in fast-moving fluid in a region are not bound to behave as fast-moving in
another region). We separate two regions in our setup, and identify three E-Curves. These
correspond to curves for the reactor, the plumbing (including filters, instrument plumbing, and the
instrument detector chamber), and the two together. Respectively, we denote them as $E_0(t)$, $E_1(t)$,
and $E_2(t)$. We are able to accurately measure $E_2(t)$ and $E_1(t)$, but not $E_0(t)$. Thus, Eq. (6) now
takes the form

$$E_2(t) = E_0(t) * E_1(t), \qquad (7)$$



and we need to solve for $E_0(t)$. Details of the deconvolution approach can be found in Appendix
B, however direct application of this technique failed to get the solution to converge. It is a robust
protocol to accurately determine a numerical RTD, and should be applied whenever a stable
solution is available.

What we propose is an indirect application, i.e., to guess $E_0(t)$ so that the convolution integral
yields a curve that matches that of $E_2(t)$. This requires a formidable number of guesses and
iterations and could be a lengthy process if done numerically. One workaround is to assume a form
of $E_0(t)$, ideally with one variable parameter, that can be tuned to give the $E_2(t)$ that best matches
the experimental $E_2(t)$ curve. The CSTR and PFR forms should not be considered since they are
ideal extremes of reactor behavior. We chose to apply the tank-in-series (TIS) model (MacMullin
and Weber Jr., 1935) to the convolution integral since it is a one parameter model that, although
not specific to flowtube, tubular, laminar, or plug-flow reactors, gives an idea of where the reactor
lies on the spectrum of mixed flow vs. plugged flow based on the value of a parameter, $N$. $N$ refers
to the fictitious number of equivalent CSTRs that, in series, describe the E-Curve for the reactor.
This function is

$$E(t) = \frac{t^{N-1}}{(N-1)!\left(\frac{\bar{t}}{N}\right)^N} e^{-\left(\frac{N}{\bar{t}}\right)t} \tag{8}$$

$$E(\theta) = \frac{N(N\theta)^{N-1}}{(N-1)!} e^{-N\theta}. \tag{9}$$




For a value of $N = 1$, the E-Curve becomes that of a perfect CSTR; for a value of $N = $ infinity, it
becomes that of a perfect PFR, as shown in Fig. S2. Using this model, the convolution integral
takes the form

$$E_2^*(t) = \int_0^t E_1(t - t') \cdot \frac{t'^{N-1}}{(N-1)!\left(\frac{\bar{t}}{N}\right)^N} e^{-\left(\frac{N}{\bar{t}}\right)t'} dt', \qquad (10)$$

where $E_1(t - t')$ is an array of accurate experimental data already obtained, and $E_2^*(t)$ is the
output guess. $E_2^*(t)$ is then matched to $E_2(t)$ by varying $N$ in an iterative fashion. Using this form,
the algorithm in Appendix B is still valid. We used MATLAB to solve this for all cases. The results
are displayed in Fig. 3.

**4 Discussion**

The small aspect ratio of the WU-PAM limits wall interactions, preventing laminar flow
development due to absence of a boundary layer. This suggests the flow field would then depend
on inlet/outlet geometries or volumetric flowrate. Though, for a fixed spacetime of 78s, we
observed that different configurations had no significant effect on the RTD (Figs. 3b, d, e). Further,
for configuration I, different spacetimes also had no significant effect. The only case with a marked
change in the signal was for configuration IV at 411s spacetime (Fig. 3f). We attribute this
difference to the low volumetric flowrate, implying that advective transport begins to be less
dominant than turbulent or molecular diffusivity as mode of transport. Such a low spacetime, while
increasing the degree of plug flow, would result in a potentially significant loss of semivolatile or
low volatility gases. Additionally, other modes of transport such as convective effects (vertical
mixing for non-isothermal conditions) could become more apparent, as revealed by Huang et al.
for the Caltech photooxidation flow tube (CPOT) reactor. As mentioned earlier, a detailed



phenomenological modeling study of RTDs in the WU-PAM is beyond the scope of this study,
however at more conventional spacetimes, it would be helpful to visualize hydrodynamics to
assess what contacting patterns and state of mixing the reactor exhibits. We thus chose a simple
scenario as a base case for simulation: configuration I at 78s spacetime.

CFD reveals that the hydrodynamics inside the PAM are far from that of a well-mixed reactor (Fig.
4). This is insightful because the F-Curve of the simulation matches reasonably well with that of
the experiment (Fig. 5) and alone would imply CSTR-like mixing. This is the caveat associated
with interpreting RTDs, and further supports investigation in phenomenological modeling.
Snapshots of the simulation displayed in Fig. 4a-c show there is jetting (short-circuiting),
recirculation, and dead zones. Jetting leads to fluid elements that have a very short residence time
and cause high values of E(t) at t > 0s. Recirculation leads to fluid elements spending more time
in the reactor, yielding middle values of E(t) as elements exit at t ~ $\bar{t}$. Stagnation (dead zones) at
the inlet of the reactor cause fluid elements to remain entrained in the reactor for a long time before
exiting the reactor at ~ 2-3 times $\bar{t}$ at low values of E(t), leading to a long tail in the E-Curve. These
three effects together lead to an E-Curve that looks similar to that of a CSTR, but mixing in CSTRs
is dominated by recirculation; meaning that the local concentration of tracer at the exit is identical
to all other locations in the reactor (Zwietering, 1959). Therefore, while tracer tests give a general
idea about contacting patterns, CFD visualizes the hydrodynamics, and help model the reactor.
Plotting the WU-PAM OFR's E-Curves for this scenario on a semilog plot does not yield different
gradients, which would otherwise indicate different volumes for the compartmental modeling of
the jetting, recirculation, and dead volumes (Levenspiel, 2002). The limitation to that statement is
that the E-Curves in this work have been obtained by fitting a one-parameter model, consequences





of which should be the focus of future work in conjunction with phenomenological modeling.
Furthermore, our simulations are limited to isothermal conditions, therefore cannot predict
buoyancy effects that could explain spread in the RTD at low flowrates (or low Reynolds numbers)
(Fig. 3f), as observed by Huang et al. (2016).

Lambe et al. (2011a) modeled the Pennsylvania State University PAM (PSU-PAM) reactor using
a compartmental model consisting of two parallel tubular reactors that exhibit Taylor dispersion
(Taylor, 1953), suggesting that their reactor (whose geometry is identical to that of the WU-PAM
OFR) has two main volumes: an active reactor volume, and another volume with entrainment. The
model output matches their experimental data reasonably well, but, they did not decouple the
reactor's E-Curve from that of the setup, implying the match may include phenomena occurring
in other pipes of the setup. Lambe et al. describe RTDs for the two volumes using the axial
dispersion model (ADM) (Taylor, 1953, 1954a, 1954b), which is based on modeling plug or
laminar flow with axial dispersion of material. Generally, as also stated by Huang et al. (2016),
the ADM is valid for regions where the radial Péclet number ($Pé_r$) is less than ~4 times the aspect
ratio (length of reactor divided by its cross sectional area),  or if $Pé_r$ is greater than $\sqrt{48}$ (Aris, 1956;
Taylor, 1954b). Both the PSU-PAM OFR and the WU-PAM OFR meet these requirements under
typical flowrates (see SI, Sect. S4). If the reactor could be described by the ADM, CFD would
show that the entrance and exit effects would be separate from the main flow in the tube – which
is not the case for the simplified geometry of configuration I. We do not know how well they apply
to the other configurations. At no point inside the reactor does pipe flow fully develop, so the high
aspect ratio concept (Kang et al., 2007) does not allow a velocity profile to become established
with the current end caps used. Thus, although $Pé_r$ appears acceptable, the inlet and outlet regions





should be re-engineered to allow formation of fully developed pipe flow in the main cylinder for
the ADM to be valid. While the E-Curve for configuration II is similar to that of configuration I at
78s spacetime, it would be helpful to run CFD on that configuration at different spacetimes to
observe if, and if so at what spacetime, the sparger and baffles efficiently suppress jetting.
Unfortunately, our CFD mesh could not be refined enough to capture the geometry of those without
sacrificing valuable computational time.

Instead, we chose to apply the use of an inlet cone (45° angle, 4.94'' length) and outlet peripherals
to simulate a more attenuated inlet and exit from sudden aperture. The results are displayed in Fig.
6. While the size of the jet appears to be broader compared to simulations in Fig. 5 (unaltered
PAM geometry), it is nonetheless present. Furthermore, recirculation in the form of backmixing is
evident towards the front, and stagnation close to the walls and corners persists. From the velocity
field (Fig. 6 center figure), a smaller cone angle that follows the contour of the light blue velocity
field could prevent backmixing.

**5 Potential implications**

Recent modeling work assumes plug flow behavior in OFRs (Li et al., 2015; Peng et al., 2015,
2016). Li et al. state that correcting for the non-ideal E-Curve in their OFR would account for
~10% error in their results, which is less than the overall model uncertainty. However, for
compounds with low $OHR_{ext}$, contacting could influence the model results to a greater extent. By
taking a ratio of characteristic reaction time (e.g., $OHR_{ext}$) to the characteristic transport time, one
can define the Damköhler number ($Da_n$). Considering spacetimes of 52-411s (as per this study),





the value of $Da_n$ can be between 5200 and 41100 for a compound with $OHR_{ext}$ ~100s$^{-1}$. Since
reaction timescales are $10^4$ times faster than transport timescales, contact patterns won't matter to
a large degree. However, the value of $Da_n$ can be between 5.2 and 41.1 for a compound with
$OHR_{ext}$ ~0.1s$^{-1}$, in which case contacting patterns may play a more significant role. This could be
the case for heterogenous reactions, diffusion-limited reactions, or semivolatile compound
(SVOC) oxidation that exhibit slow gas-particle partitioning. Furthermore, combining
phenomenological model to an associated RTD can impact kinetics (and yields) further. The RTD
generated by Lambe et al. (2011) employed in Li et al. (2015) may lead to greater than 10% error
if the 2 PFRs in parallel model suggested by Lambe et al. (2011) is not applicable. In these
scenarios, ensuring a high degree of plug flow can not only maximize exposure, but minimize the
distribution of aged compounds (e.g., first or second generation compounds) that are due to
different exit ages because of recirculation or stagnation. This configuration would suit a
laboratory experiment with slow kinetics, where concentrations can be made high enough to where
wall losses aren't an issue. However, this configuration may not suit a field deployment where
trace compounds have high $OHR_{ext}$ and can be easily lost to reactor walls, in which case ensuring
a high degree of mixing would be beneficial.

**6 Conclusion**

The WU-PAM reactor's hydrodynamics are complex, and even though the E-Curve looks simple,
applying a compartmental model (phenomenological modeling) to obtain an analytical E-Curve
(rather than the empirically-based TIS E-Curve) can be challenging. Having too sudden an aperture
at the entrance zone leads to dead volumes at the inlet corners. We cannot confirm if the sparger

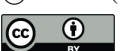



design helps reduce dead volume, but tracer tests suggest it doesn't appear to affect the degree of
plug flow under standard operating spacetimes (52-156s). The reactor is described neither by back
mixing, plug flow, nor by the ADM in any configuration. However, for configuration IV at 411s
spacetime, a noticeable shift towards plug flow behavior is observed, perhaps due to a combined
effect of internals and low inlet velocity. We note that the E-Curves we obtain are not as accurate
as an E-Curve numerically obtained by direct deconvolution, since we are forcing a closed form
solution on our data. We further note the need for phenomenological modeling.

Tapered ends on the inlet and the outlet would help to develop a steady flow profile at the inlet
and avoid recirculation at the outlet, however the cone angle should be predetermined by CFD if
possible. By improving simulations to include temperature gradients induced when the internal
lamps are on, and refining the mesh to capture internals, the ADM should be revisited as a model
to describe the PAM reactor. If the ADM satisfactorily describes the PAM reactor's RTD, kinetics
should be easier to obtain, and diffusivity values using the Aris-Taylor relationship (Aris, 1956)
can even be obtained. This could help assess whether processes are reaction limited or diffusion
limited, arguing the reactor validity in experimental setups. At that point, the reactors would be
regulated by only one parameter, their flowrate. This parameter would be adjusted to achieve
desired spacetimes depending on $OHR_{ext}$. Finally, to obtain accurate experimental RTDs,
achieving a functional direct deconvolution code should be focus of future development. The
implementation of this technique can be extended to drift tubes in mass spectrometers, as those are
essentially flow tube reactors where ionization efficiency can be strongly influenced by mixing.

**Acknowledgements**




We would like to express appreciation for the valuable discussions with Prof. Jay Turner, Prof.
James Ballard, Christopher Oxford, David Hagan, and Tim Lee at Washington University in St.
Louis, and valuable correspondence with Prof. William Brune at the Pennsylvania State University
and Dr. Andrew Lambe at Aerodyne Research Inc. We would also like to thank ENGYS and Prof.
Milorad Duduković's CREL resources, who provided the necessary computational power to run
CFD. This work was partly funded by the National Science Foundation (NSF) CBET Award
#1236865, and NSF CBET Award #1437933.

**Figures**





Figure 1: Experimental setup for tracer studies for a) one inlet and one outlet and b) peripheral
inlets and outlets. The main difference is the presence of the ring sparger in b).





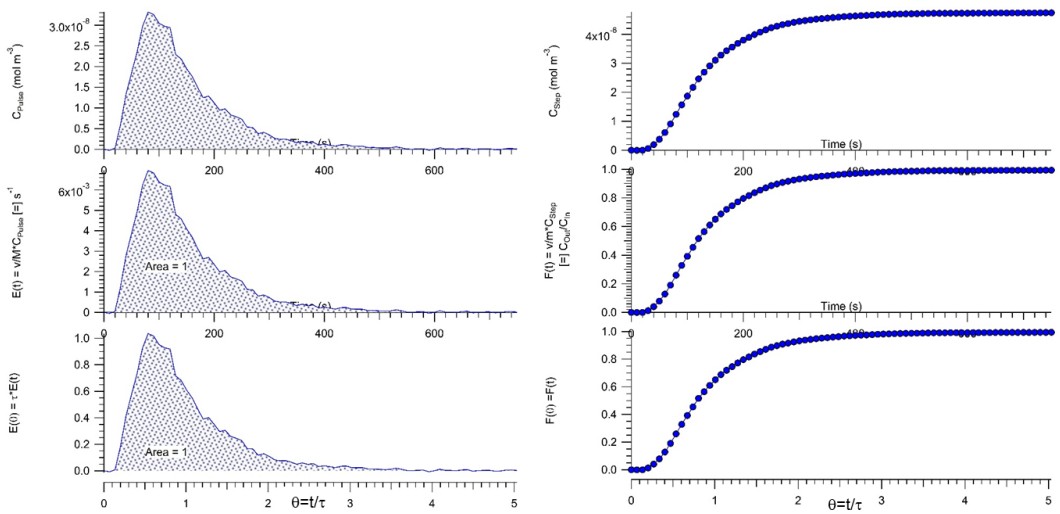

Figure 2: Tracer tests at 10 L min$^{-1}$ (78s spacetime) through the reactor for configuration I. This figure serves as an illustrative example for non-dimensionalizing tracer response curves.

Figure 3: E-Curves for the WU-PAM configuration I at a) 52s b) 78s c) 156s spacetimes, at 78s
spacetimes for d) configuration II e) configuration III, and f) for configuration IV at 411s
spacetime. Details on the configurations are in the Methods section.



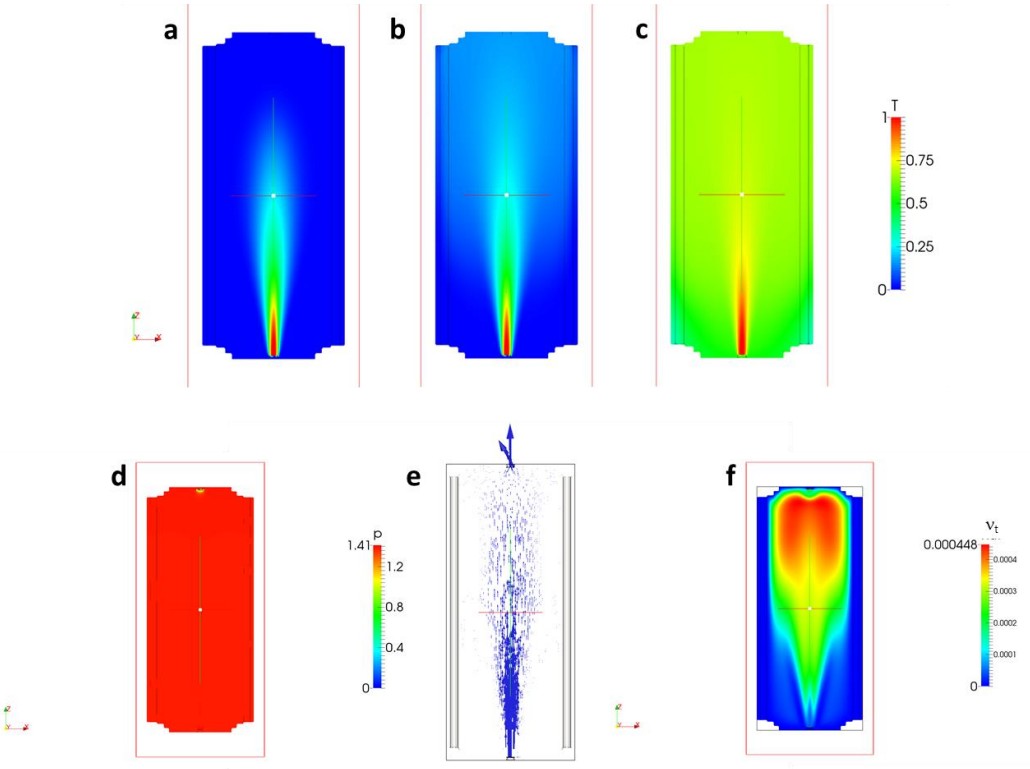

Figure 4: CFD output for configuration I at 78s spacetime: snapshots at a) 1s b) 10s and c) 100s of
runtime, and d) pressure field, e) velocity field, and f) turbulent diffusivity field. Color scales are
dimensionless scalar concentration for the tracer (a-c), Bar for the pressure field (d), and cSt for
the kinematic viscosity (f).





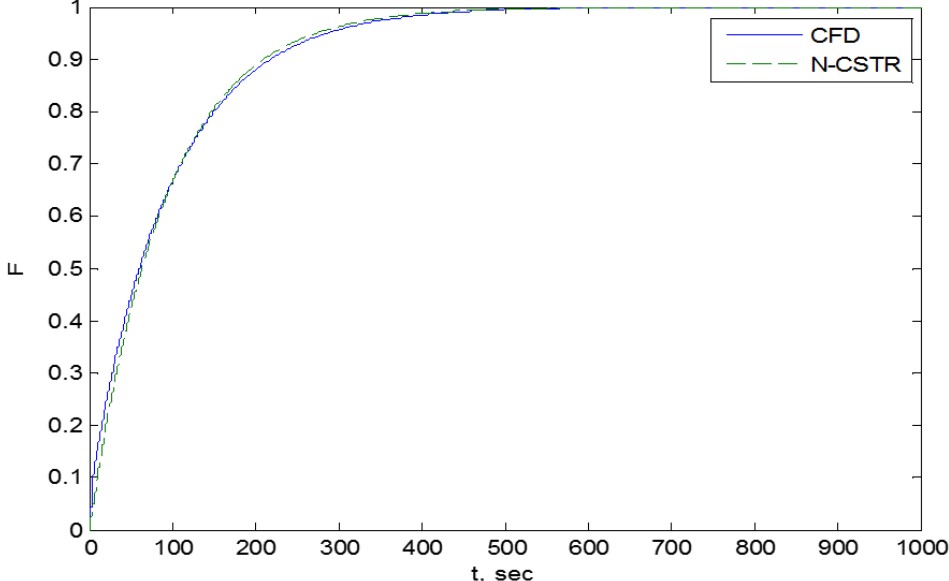

Figure 5: Comparison of F-Curve output between CFD and tracer test for configuration I at 78s spacetime.

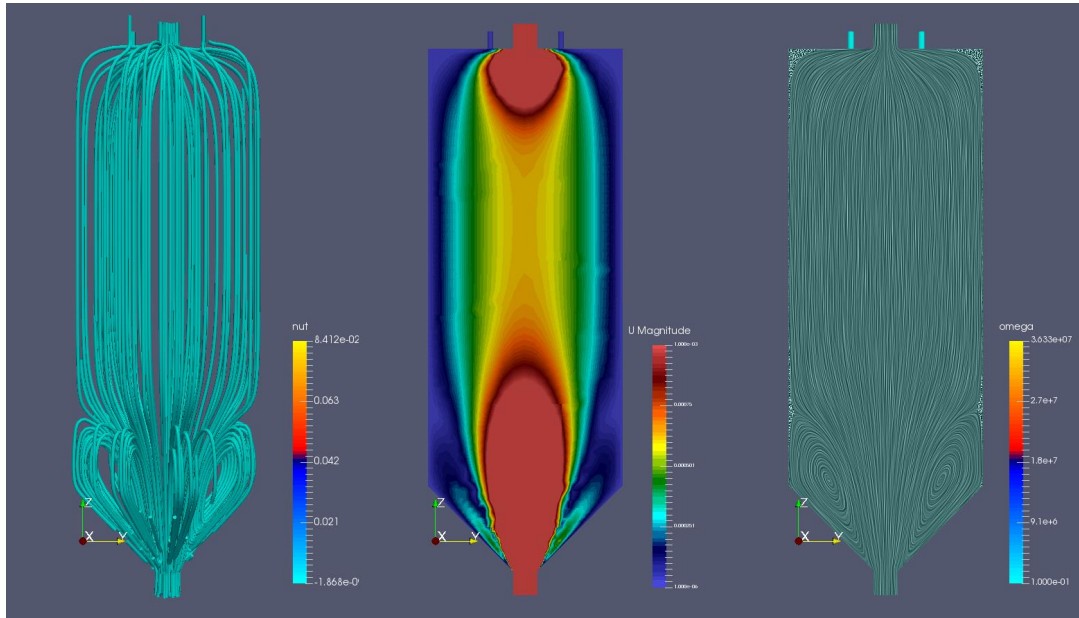

Figure 6: CFD analysis on the effect of inlet cone and peripheral outlets on fluid flow. All figures represent a visualization of the flow field, with color scales representing (from left to right): kinematic viscosity, velocity, and $\omega$. The 3D representation on the leftmost figure highlights the uniformity of the recirculation region.





## Appendix A: The use of E-Curves and F-Curves

To determine RTDs, we injected tracer in a steady stream rather than a single pulse. This prolonged and constant injection, which we call a step input, gave us F(t), from which we can derive E(t) as follows:

$$F(t) = \frac{v}{m} C_{step} \tag{A1}$$

$$E(t) = \frac{dF(t)}{dt}, \tag{A2}$$

where $v$ is the volumetric flowrate in $m^3 s^{-1}$, $m$ is the molar flowrate of the tracer in mol s$^{-1}$, and $C_{step}$ is the concentration of the tracer for a step input in mol m$^{-3}$. Therefore, F(t) is dimensionless, and E(t) in this example has units of s$^{-1}$. The area under the E-Curve is unity, representing the PDF of the system:

$$\int_0^\infty E(t)dt = 1. \tag{A3}$$

Similarly, for the dimensionless domain

$$\int_0^\infty E(\theta)d\theta = 1. \tag{A4}$$

And if we take $\bar{t}$ to be the mean residence time of the reactor, then

$$\theta = \frac{t}{\bar{t}}. \tag{A5}$$

The additional utility of the dimensionless domain is that for reactors of different sizes, built to behave the same, the RTD is numerically identical. For example, if PAM OFRs are operated in different ways (e.g., they operate at different flowrates) or are built in different sizes but display the same E-Curve in the dimensionless domain, then their performance will be identical, and their mean residence time will always occur at θ = 1. This identity would apply for the F-Curve as well in both domains, where from Eq. (A2) we can see that




$$F(t) = \int_0^t E(t)dt \qquad \text{(A6)}$$

$$F(\theta) = \int_0^t E(\theta)d\theta. \qquad \text{(A7)}$$

The mathematical properties of interest for PDFs are their moments: These have quantitative
meanings in E-Curve analysis. A general equation for the moments of a function $f(x)$ is

$$\mu_n = \int_{-\infty}^{\infty} x^n \cdot f(x)dx, \qquad \text{(A8)}$$

where $\mu_n$ is the nth moment of the distribution. If we consider a raw C(t) dataset from our tracer,
we can derive the moments:

$$\frac{\int_0^\infty C(t)dt}{\int_0^\infty C(t)dt} = \int_0^\infty E(t)dt = 1 = \mu_0 \qquad \text{(A9)}$$

$$\frac{\int_0^\infty t\cdot C(t)dt}{\int_0^\infty C(t)dt} = \int_0^\infty t \cdot E(t)dt = \bar{t} = \mu_1. \qquad \text{(A10)}$$

Here, we are interested in the first moment, which represents the mean residence time. For higher
moments, we use the central moments of the distribution since we are interested in quantities like
variance, skewness, and kurtosis around the mean (and not around zero). This alters Eq. (A8) as
follows:

$$\mu_n = \int_{-\infty}^{\infty}(x - a)^n \cdot f(x)dx; n \geq 2, \qquad \text{(A11)}$$

where $a$ is a constant, and is generally the mean of the distribution ($\bar{t}$ in this case). Thus, the second
(central) moment of the E-Curve becomes

$$\frac{\int_0^\infty (t-\bar{t})^2 \cdot C(t)dt}{\int_0^\infty C(t)dt} = \int_0^\infty (t - \bar{t})^2 \cdot E(t)dt = \sigma^2 = $$

$$\mu_2, \qquad \text{(A12)}$$



where $\sigma^2$ has a clear physical meaning, and is the variance around the mean. Higher moments
(skewness and kurtosis) can be of use, and require additional math, but are not addressed in this
work.

**Appendix B: Algorithm for direct deconvolution**

Here, we perform an inverse operation to Eq. (7) (Sun, 2010) and work towards an output curve:

$$E_2(t) = \int_0^t E_1(t - t') \cdot E_0(t')dt'. \tag{B1}$$

In discrete form, taking a constant time step $\Delta t$, we can take a datapoint at $t_i = i\Delta t$,

$$E_{2_N}(t_i) = \sum_{i=1}^{N} \int_{t_{i-1}}^{t_i} E_1(t_N - t') \cdot E_0(t')dt'. \tag{B2}$$

If we then assume that the functions $E_1(t - t')$ and $E_0(t')$ are constant for the interval $t_{i-1} \leq t' \leq$
$t_i$, we can simplify the integral:

$$E_1(t_N - t') = \frac{1}{2}(E_1|_{N-1} + E_1|_{N-i+1}) \tag{B3}$$

$$E_0(t') = \frac{1}{2}(E_0|_i + E_0|_{i-1}) \tag{B4}$$

$$\int_{t_{i-1}}^{t_i} E_1(t_N - t') \cdot E_0(t')dt' = \frac{1}{4}(E_1|_{N-1} +$$

$$E_1|_{N-i+1})(E_0|_i + E_0|_{i-1})\Delta t. \tag{B5}$$

Now, Eq. (B2) becomes

$$E_{2_N}(t_i) = \sum_{i=1}^{N} \frac{1}{4}(E_1|_{N-1} + E_1|_{N-i+1})(E_0|_i +$$

$$E_0|_{i-1})\Delta t. \tag{B6}$$





From experimental data, we can accurately collect datapoints for $(E_1|_{N-1} + E_1|_{N-i+1})$ as well as
$E_2(t_i)$, so we need to rearrange for $(E_0|_i + E_0|_{i-1})$, which has to be solved numerically in matrix
form. Let

$$\alpha_i = \beta_i = \frac{1}{4}(E_1|_{N-1} + E_1|_{N-i+1}) \tag{B7}$$

$$A_{N,i} = \begin{cases} \alpha_i + \beta_{i+1} & i = 1,2,\dots,(N-1) \\ \alpha_N & i = N \end{cases}. \tag{B8}$$

Upon the initial condition

$$B_N = \frac{E_2(t_N)}{\Delta t} - \beta_1 E_0|_0, \tag{B9}$$

we have that

$$B_N = \sum_{i=1}^{N} A_{N,i} E_0|_i. \tag{B10}$$

Finally,

$$\overrightarrow{E_0(t)} = \vec{A}^{-1}\vec{B}. \tag{B11}$$

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
