# Peer review of "Assessing the degree of plug flow in oxidation flow reactors (OFRs): a study on a Potential Aerosol Mass (PAM) reactor"

_Atmospheric Measurement Techniques, 2017_

## Referee Comment (RC1) · Anonymous Referee #2 · 21 Dec 2017

This manuscript describes a modeling-measurement comparison of residence time distributions in a PAM-style oxidation flow reactor (OFR). The authors show that under their experimental setup determining the true reactor RTD requires deconvolution of the plumbing external to the reactor. It is shown that for various changes in the gas introduction and exit configuration (single tube, sparger, rings), the RTD is indistinguishable. A CFD simulation shows that despite the RTD suggesting a well-stirred reactor, the flow pattern is quite different with central jetting, recirculation and dead zones. An additional CFD simulation showed that with a cone on the inlet, the flow pattern is similar and not plug-flow.

The paper is generally well-written and explores an understudied and important aspect of OFRs which are rapidly becoming more widely used in atmospheric lab and field studies. Therefore, I recommend publication after a few minor-to-moderate revisions described below.

**General/Main Comments:**

- The authors seemed to have missed some very relevant recently-published work on RTDs and some of the effects on kinetics and VOC reactions (see details below) which provide additional context for framing this work.

- The "Potential Impacts" section could use substantial improvement. The OH reactivity (OHRext) usage seems inaccurate and the discussion following is therefore unclear (see details below). Also, this section seems a bit abstract and underdeveloped, in that it doesn't convey how these issues may actually impact real experiments and applications that people are using OFRs for. I would recommended framing and expanding the points made to discuss how they might affect results for typical applications. I.e. SOA yields or compositions, gas-product formation, aerosol chemical or physicochemical transformation (e.g. hygroscopicity), etc. Given that the manuscript is intended for publication in an atmospheric-focused paper, a stronger connection to how this study will help advance measurements related to the atmosphere is important.

- Indeed, the experimental setup used in this study requires backing out the substantial delays and smearing of the gas delivery and measurement systems. The authors do a nice job of working out and explaining a method to accurately extract the true OFR RTDs. However, in practice wouldn't it be best to minimize the plumbing and/or detection delays using a system with a much faster response time? E.g. use of 1/8" tubing, CO2 gas, and LICOR CO2 detector could achieve response times of probably only a few seconds, couldn't it? It would be useful to discuss/recommend the best experimental practices to most easily and accurately extract the parameters that other OFR users could then apply to their systems, based on what was learned in this study.

[Figure]

**Detailed Comments:**

- P5, L97: Ortega et al. 2015 should be updated to ACP 2016.

- P5, L97-104 (and latter part of the intro such as P6, L126-128) seems to be missing some of the recent literature related to measured/modeled RTDs and chemical effects in PAM-type OFRs that would provide better context of what has/hasn't been done in terms of modeling/characterizing OFR flow (especially PAM-type most relevant to this work). These include (but may not be limited to): Peng et al. 2015 (which the authors cite earlier) expands substantially on the Li et al paper and discusses how different flow RTD assumptions (plug, laminar, measured) affect OH exposure (see Section 3.5, Figs. 9, 10, S11, S12, S16, Table S1); Ortega et al. ACP 2016 (cited elsewhere) shows FLUENT CFDs of with/without the inlet plate installed (Section 2.2, Fig. S1); Palm et al. ACP 2017 (www.atmos-chem-phys.net/17/5331/2017/) shows RTDs from FLUENT CFDs without the inlet plate installed for the PAM OFR for different particles sizes and compared to the Lambe et al 2011 RTD. (Section 2.2, Fig. S1); Palm et al. ACPD, 2017 (https://doi.org/10.5194/acp-2017-795) shows some modeled chemical differences (in VOC decays) for different RTD flow assumptions (Figs. 1, 2, S6).

- P7, L161: consider reporting SO2 tank concentration.

- P8, L182: delete "create" or "allow"

- P10, L218: add hyphen for first-order

- P10, L224-225: What is meant by "however those simulations required significant computer time to resolve mesh sizing"? Do the authors mean to say it would take too much time to run (or justify running) for this study?

- P18, L395-397: Again, more detailed modeling work from other publications on effects of differing RTDs and flow assumptions missing here.

- P18-19: "Potential Implications" section. The use of OHRext appears to not be accurately used. OHR is not an exclusively intensive property of a compound (as seems to

be implied in the text) but rather depends on the concentration and OH rate constants of the compounds present that can react with OH. Also OHR is a measure of the (inverse) OH lifetime, not its reaction partners. Maybe the authors really mean the OH lifetime of different compounds? i.e. kvoc+oh x [OH].

- P19, L412-16: It's not clear why compounds that react faster with OH would be more prone to be lost to the reactor walls. It seems that the opposite is stated above. Also not clear how rapid mixing would help that situation.

- P19, L406-407. Add "a" before phenomenological or make "model" plural.

- P19, L412: Statement: "This configuration would suit a laboratory experiment with slow kinetics, where concentrations can be made high enough to where wall losses aren't an issue." This statement may be very misleading. Simply increasing concentrations in many cases does not decrease the relative importance of wall effects since they are often first-order losses and the walls may not necessarily establish equilibrium and relevant timescales. Please revise to precisely state what is meant here, or possibly delete if not relevant.

- P20, L442: add "the" or "a" before "focus"

Figures:

- Fig. 1: Higher resolution on detailed photos needed. This may have just been the pdf conversion that shouldn't be an issue if high-resolution pictures provided for final publication. Otherwise, the thorough photographic documentation is a nice inclusion.

- Fig. 1b "internals" photo: black label too hard to read on dark background. Try white or yellow and move to the right.

- Fig. 2: all text too small (axes labels, tick label). Also x-axes labels on two plots on left are hiding behind data

- Fig. 4. Units for velocity missing. Also, the colorbar and labels are too small.

[Figure]

---

## Referee Comment (RC2) · Anonymous Referee #1 · 25 Dec 2017

Mitroo et al. introduce a method to deconvolve the measured residence time distribution (RTD) from sampling tubes to get the real RTD inside the Washington University Potential Aerosol Mass (WU-PAM) reactor, which is also validated by computational fluid dynamic (CFD) simulation. The idea of this paper can help improve the understanding of RTD for the oxidative flow reactor (OFR) user community. This paper is well-written and fits the scope of AMT. I suggest for publication after considering the following aspects:

General comments:

1. I agree with Review #2's comments about the expansion of Section "5 Potential im-

plication". My concern is that how your method can be applied to simulation rather than just used to explain RTD. In other words, how does the incorporation of CSTR tank-in-series (TIS) model framework behave when compared with the PFR framework? For example, most of your inversion results indicate that the number of TIS, N, is a little bit larger than 1. Does that mean it is CSTR rather than PFR that can better represent OFR? So to simulate what happens in OFR, we should use CSTR model instead of PFR? Then the question is to what extent the difference will be introduced to the simulated results by shifting from PFR to CSTR. I think the authors should clarify these points in this section.

2. TIS model can have different forms. The authors assume the same residence time for each CSTR-tank and find the tank number. One can also take the form with a fixed CSTR (or PFR or mixed CSTR/PFR) number but to find each residence time, which looks more reasonable given the CFD simulation. Can the authors discuss this a little bit more? For example: How does the number of TIS, N, depend on the flow rate, or in other words the average residence time? Since the flow rate changes the fluid field, the mixing style could be different at different flow rates (e.g. Fig.3a-c). But I cannot see any trend. Can the author give some explanation for that?

3. Equations in Appendix B should be carefully checked. For example, in Eq. (B3) it should be $E_1|_{N-i}$ instead of $E_1|_{N-1}$. In addition, try to avoid "N", since the number of TIS is also "N", which may cause confusion. In Eq. (B8), **A** is a matrix, which should be listed as $A_{i,j}$ not just $A_{N,i}$. One more, as a vector, **B** should be listed as $B_i$, with $i = 1, 2, ..., N-1$ and $i = N$. About time step $\Delta t$, see following comments for Figure 3.

Specific comments:

1. Line 182: "create allow", delete either one.

2. Line 218: "F-curve", define it here or mention it later.

3. It is unnecessary to list both dimensional and non-dimensional equations at the

same time, e.g. EQ. 1-4 and 8-9, since the non-dimensional form has been introduced in detail in Appendix A.

4. Figure 2: Please use higher resolution figures and rearrange the figure locations (too compact, and x-labels are hidden).

5. Figure 3: Why time resolution is different in Panel e? Does $\Delta t$ in Appendix B correspond to the time interval in Figures 3a-f?

6. Figure 5: Please use an intuitive y-label instead of "F". Also please specify "N" value in the caption.

---

## Author Comment (AC1) · 22 Feb 2018

**RESPONSE TO REVIEWER 1: Manuscript ID: amt-2017-352**

*This manuscript describes a modeling-measurement comparison of residence time distributions in a PAM-style oxidation flow reactor (OFR). The authors show that under their experimental setup determining the true reactor RTD requires deconvolution of the plumbing external to the reactor. It is shown that for various changes in the gas introduction and exit configuration (single tube, sparger, rings), the RTD is indistinguishable. A CFD simulation shows that despite the RTD suggesting a well-stirred reactor, the flow pattern is quite different with central jetting, recirculation and dead zones. An additional CFD simulation showed that with a cone on the inlet, the flow pattern is similar and not plug-flow. The paper is generally well-written and explores an understudied and important aspect of OFRs which are rapidly becoming more widely used in atmospheric lab and field studies. Therefore, I recommend publication after a few minor-to-moderate revisions described below.*

We thank the referee for their time in reading and reviewing this manuscript.

**[In reference to: Main Comments]**

*- The authors seemed to have missed some very relevant recently-published work on RTDs and some of the effects on kinetics and VOC reactions (see details below) which provide additional context for framing this work.*

We thank the referee for pointing this out. Upon review of (Palm et al., 2017, 2018; Peng et al., 2017; Peng and Jimenez, 2017) we restructured our arguments in the 'Introduction' section, as well as in the 'Potential Implications' section. Please see our responses to the detailed comments, kindly offered by the referee, below.

*- The "Potential Impacts" section could use substantial improvement. The OH reactivity (OHRext) usage seems inaccurate and the discussion following is therefore unclear (see details below). Also, this section seems a bit abstract and underdeveloped, in that it doesn't convey how these issues may actually impact real experiments and applications that people are using OFRs for. I would recommended framing and expanding the points made to discuss how they might affect results for typical applications. I.e. SOA yields or compositions, gas-product formation, aerosol chemical or physicochemical transformation (e.g. hygroscopicity), etc. Given that the manuscript is intended for publication in an atmospheric-focused paper, a stronger connection to how this study will help advance measurements related to the atmosphere is important.*

We thank the referee for suggesting a more concrete point of view for the implications of our work. We have done our best to address this, and with the review of the recent literature, believe the 'Potential Implications' section is significantly improved as a result. We have included excerpts, in quotations, of new sections as well as improved wording of the revised manuscript in response to the Detailed Comment section, below.

*- Indeed, the experimental setup used in this study requires backing out the substantial delays and smearing of the gas delivery and measurement systems. The authors do a nice job of working out and explaining a method to accurately extract the true OFR RTDs. However, in practice wouldn't*

*it be best to minimize the plumbing and/or detection delays using a system with a much faster response time? E.g. use of 1/8" tubing, CO2 gas, and LICOR CO2 detector could achieve response times of probably only a few seconds, couldn't it? It would be useful to discuss/recommend the best experimental practices to most easily and accurately extract the parameters that other OFR users could then apply to their systems, based on what was learned in this study.*

> The referee offers a fair, thought-provoking argument. On one hand, an alternate method to minimize plumbing and increase response time may minimize bias in the results (although can still be present). On the other hand, our method entirely removes operation-specific bias, and is conveniently adaptable to multiple inlets and outlets, or any other operation-specific arrangement that may be required given other constraints. We believe there are multiple scenarios in which minimizing plumbing and utilizing on-hand instrumentation, such as the LI-COR LI-820 $CO_2$ Analyzer, may not be feasible or cost-effective. For example, there can be physical/space limitations as to how short plumbing lines can be, there may be a need for peripheral inlets and outlets (e.g., "cNO" idealized configuration in (Peng et al., 2017)) or requirements for additional dilutors, scrubbers, etc.,

> Also, novel techniques strive to minimize residence time in OFRs (e.g., Simonen et al., 2016). It is arguable if in exceedingly large OFRs, such as the CPOT (Huang et al., 2016), the spacetime in the lines is so small that any RTD in the plumbing will not affect the overall RTD. However this assumption loses validity as the spacetimes of the tubing and reactor become comparable or non-negligible. The RTD should only apply for fluid moving in the reaction zone – and for OFRs the reaction zone is confined to the zone illuminated by the UV lamps (the OFR itself). Our method also allows for data correction post-experiment, if necessary. While we are hesitant to offer best practice recommendations without quantitative data, we have included the following (lines 454-461 of the revised manuscript):

> "We do recognize that OFR (or any environmental chemical reactor) users may have a preference to rapidly obtain an RTD profile perhaps using an improvised setup with very short sample lines and a fast time-response gas analyzer. However, the accuracy to which the profile is obtained should be carefully examined. If the reactor is considerably large, or if it is an OFR to be deployed for low levels of exposure, then the influence of plumbing is minimal. If the reactor of choice is small, the oxidant exposure is high, or the reactor has more than one inlet/outlet or other peripheral components, it would be recommended to use the method described here to obtain the most representative RTD, since all sources of bias are removed."

**[In reference to: Detailed Comments]**

*- P5, L97: Ortega et al. 2015 should be updated to ACP 2016.*

> We have updated the citation from the discussion article to the final article.

*- P5, L97-104 (and latter part of the intro such as P6, L126-128) seems to be missing some of the recent literature related to measured/modeled RTDs and chemical effects in PAM-type OFRs that*

*would provide better context of what has/hasn't been done in terms of modeling/characterizing OFR flow (especially PAM-type most relevant to this work). These include (but may not be limited to): Peng et al. 2015 (which the authors cite earlier) expands substantially on the Li et al paper and discusses how different flow RTD assumptions (plug, laminar, measured) affect OH exposure (see Section 3.5, Figs. 9, 10, S11, S12, S16, Table S1); Ortega et al. ACP 2016 (cited elsewhere) shows FLUENT CFDs of with/without the inlet plate installed (Section 2.2, Fig. S1); Palm et al. ACP 2017 (www.atmos-chem-phys.net/17/5331/2017/) shows RTDs from FLUENT CFDs without the inlet plate installed for the PAM OFR for different particles sizes and compared to the Lambe et al 2011 RTD. (Section 2.2, Fig. S1); Palm et al. ACPD, 2017 (https://doi.org/10.5194/acp-2017-795) shows some modeled chemical differences (in VOC decays) for different RTD flow assumptions (Figs. 1, 2, S6).*

Similar to the first comment in the Main Comments section, we thank the reviewer for highlighting this recent work. With reference to L102-112, we included the following in place of the last sentence of the paragraph:

"Following an experimentally determined RTD (Lambe et al., 2011) in a PAM OFR, Peng et al. extend the model developed by Li et al., to include this non-ideal RTD, suggesting model disagreement at high exposures. Ortega et al. employ FLUENT to show that removal of the inlet plate (resulting in a less pronounced aperture to the reactor) significantly decreases recirculation regions; and Palm et al. then extend the simulation to show that the FLUENT-derived RTD (Palm et al., 2017) has a narrower distribution than the experimentally-derived RTD by Lambe et al. Finally, Peng and Jimenez lay an initial framework for the possibility of OFRs investigating NO chemistry (Peng and Jimenez, 2017), where initial sensitivity analysis on RTDs suggest considerable model disagreement at high exposures. The fundamental caveat in this recent work is the reliance on an accurately determined experimental RTD, that provides the basis for error analysis."

With respect to the latter part of the introduction (L126-128 of the original manuscript pointed out by the referee), we believe to have reviewed the literature appropriately. The intent of this paragraph was to compare experimental methods and experimentally-obtained RTDs, such as those in Lambe et al (2011a), Huang et al. (2016), and Simonen et al. (2016). Literature suggested by the referee employs FLUENT to either model an RTD or takes the compartmental model from Lambe et al. (2011a) to interpret their results, but does not provide an approach to experimentally obtaining RTDs, per se. To clarify this point, the sentence (lines 134-136 in the revised manuscript) now reads:

"We compare this approach to that of previous studies by Lambe et al. (2011a), Huang et al. (2016), and Simonen et al. (2016), which to the best of our knowledge are the only other studies to date that report experimentally-derived RTDs in OFRs."

Nonetheless, the review of the very recent literature indeed provided better context for our introduction, for which we thank the reviewer as our manuscript is now significantly improved.

*- P7, L161: consider reporting SO2 tank concentration.*

We added a note here that states 3 ppm $SO_2$ tank concentration.

*- P8, L182: delete "create" or "allow"*

Similar to Reviewer 2, we thank the referee for pointing this out, and have deleted "allow".

*- P10, L218: add hyphen for first-order*

We changed "first order" to "first-order".

*- P10, L224-225: What is meant by "however those simulations required significant computer time to resolve mesh sizing"? Do the authors mean to say it would take too much time to run (or justify running) for this study?*

Yes. We have deleted the ambiguous phrase, and added the new sentence below (lines 233-236 of the revised manuscript):

"However, resolving the simulation mesh size to account for these internals significantly extended the computational requirements, to the point that running these simulations was not possible on our computer system and would require a computing cluster to perform."

*- P18, L395-397: Again, more detailed modeling work from other publications on effects of differing RTDs and flow assumptions missing here.*

We thank the referee for bringing this to our attention, and have included references from (Palm et al., 2017, 2018, Peng et al., 2015, 2016; Peng and Jimenez, 2017) in this section, with additional discussion suggested by Reviewer 2. We have restructured the paragraph (lines 406-434 in the revised manuscript):

"Initial PAM modeling work assumed plug flow behavior in OFRs (Li et al., 2015). Li et al. stated that correcting for the non-ideal E-Curve in their OFR would account for ~10% error in their oxidant exposure results, which is less than the overall model uncertainty. However, recent work incorporates the effect of non-ideal RTDs on model outputs (Palm et al., 2017, 2018, Peng et al., 2015, 2016; Peng and Jimenez, 2017). Peng et al. (2015) show that for three OFR operational modes (that is, modes of different oxidant formation mechanisms denoted by 'OFR185', 'OFR254-70', and 'OFR254-7'), a comparison between model output for ideal plug flow vs. non-ideal RTDs (using the RTD experimentally obtained by Lambe et al., 2011a) for OH exposure ($OH_{exp}$) generally agree within a factor of 2 for low $OH_{exp}$; the model disagreement exacerbates at high $OH_{exp}$ beyond a factor of ~4. Peng and Jimenez then extend OFR operational modes to include N-containing chemistry (in modes referred to therein as 'OFR185-iNO', 'OFR185-7-iNO', and 'OFR185-70-iNO') where at moderate-to-high $OH_{exp}$, the deviations exacerbate significantly, although the authors argue those conditions represent unrealistic chemical pathways. It is worthwhile noting that the chemistry modeled by Peng and Jimenez may find a workaround by utilizing $N_2O$ as NO precursor (Lambe et al., 2017) rather than NO

itself, potentially minimizing RTD-related errors. Palm et al. (2018) report data from OFR field deployment where the same comparison (ideal plug flow vs. the RTD experimentally obtained by Lambe et al., 2011a) suggests RTD-related errors overpredict (for CO) or underpredict (for toluene and monoterpenes) photochemical age (that is, the ratio of OHexp to tropospheric average OH number concentrations) in the reactor, generally within a factor of 3 of model error. Considering this work employs the compartmental model RTD described by Lambe et al. (2011a), which for reasons mentioned in the previous section may not be the true PAM RTD, and given that non-ideality in RTDs affects certain OFRs more than others, implementing the method presented here to obtain a more representative reactor RTD can either help constrain error uncertainty in the models, or possibly extend the $OH_{exp}$ range in which OFRs can be operated, a reportedly nontrivial task (Palm et al., 2018). Considering our results indicate that OFRs like the WU-PAM exhibit an RTD closely matching that of an ideal CSTR, which is more well-mixed than the Lambe et al. RTD, the sensitivity analysis conducted so far could represent a lower bound for error analysis because the Lambe et al. RTD is closer to a PFR-like RTD than a CSTR-like RTD."

*- P18-19: "Potential Implications" section. The use of OHRext appears to not be accurately used. OHR is not an exclusively intensive property of a compound (as seems to be implied in the text) but rather depends on the concentration and OH rate constants of the compounds present that can react with OH. Also OHR is a measure of the (inverse) OH lifetime, not its reaction partners. Maybe the authors really mean the OH lifetime of different compounds? i.e. kvoc+oh x [OH].*

We replaced '$OH R_{ext}$' with 'lifetime to OH' and restructured the paragraph (lines 436-451 in the revised manuscript):

"For compounds with low lifetimes to OH, contacting could influence the model results to a greater extent (e.g., field deployment monoterpene decay reported by Palm et al., 2018). By taking a ratio of characteristic reaction time to the characteristic transport time, one can define the Damköhler number ($Da_n$). Considering spacetimes of 52-411s (as per this study), the value of $Da_n$ can be between 0.52 and 4.11 for a compound with lifetimes of ~100s. Since reaction timescales are on the order of transport timescales, contact patterns may play an important role, as seen in Palm et al. (2018). This could also be the case for heterogenous reactions, diffusion-limited reactions, or semivolatile compound (SVOC) oxidation that exhibit slow gas-particle partitioning. Furthermore, combining a phenomenological model to an associated RTD can impact kinetics (and yields) further. The RTD generated by Lambe et al. (2011a) employed in Li et al. (2015) may lead to greater than 10% error if the 2 PFRs in parallel model suggested by Lambe et al. (2011a) is not applicable. In these scenarios, ensuring a high degree of plug flow can not only maximize exposure, but minimize the distribution of aged compounds (e.g., first or second generation compounds) that are due to different exit ages because of recirculation or stagnation. However, this configuration may not suit a field deployment where trace compounds have short lifetimes to OH and can be easily lost to reactor walls, in which case ensuring a high degree of mixing would be beneficial."

*- P19, L412-16: It's not clear why compounds that react faster with OH would be more prone to be lost to the reactor walls. It seems that the opposite is stated above. Also not clear how rapid mixing would help that situation.*

We removed this argument from our discussion.

*- P19, L406-407. Add "a" before phenomenological or make "model" plural.*

We incorporated "a" in the sentence:

"Furthermore, combining a phenomenological model to an associated RTD can impact kinetics (and yields) further."

*- P19, L412: Statement: "This configuration would suit a laboratory experiment with slow kinetics, where concentrations can be made high enough to where wall losses aren't an issue." This statement may be very misleading. Simply increasing concentrations in many cases does not decrease the relative importance of wall effects since they are often first-order losses and the walls may not necessarily establish equilibrium and relevant timescales. Please revise to precisely state what is meant here, or possibly delete if not relevant.*

Upon re-examination of the sentence, we removed the sentence altogether.

*- P20, L442: add "the" or "a" before "focus"*

We incorporated "a" in the sentence:

"Finally, to obtain accurate experimental RTDs, achieving a functional direct deconvolution code should be a focus of future development."

*Figures:*

*- Fig. 1: Higher resolution on detailed photos needed. This may have just been the pdf conversion that shouldn't be an issue if high-resolution pictures provided for final publication. Otherwise, the thorough photographic documentation is a nice inclusion.*

We will work with the editor to ensure high quality images.

*- Fig. 1b "internals" photo: black label too hard to read on dark background. Try white or yellow and move to the right.*

We moved "internals" to the center of the insert, and changed the font color to white.

*- Fig. 2: all text too small (axes labels, tick label). Also x-axes labels on two plots on left are hiding behind data*

We restructured Fig. 2 accordingly, incorporating requests from Reviewer 2 as well.

- Fig. 4. Units for velocity missing. Also, the colorbar and labels are too small.

> Insert (e) represents a vector field for the velocity, so units are not needed. We stated this in the caption for clarification. The legend size cannot be changed as it comes out of OpenFOAM this way.

**References**

Huang, Y., Coggon, M. M., Zhao, R., Lignell, H., Bauer, M. U., Flagan, R. C. and Seinfeld, J. H.: The Caltech Photooxidation Flow Tube Reactor – I: Design and Fluid Dynamics, Atmospheric Meas. Tech. Discuss., 1–36, doi:10.5194/amt-2016-282, 2016.

Lambe, A., Massoli, P., Zhang, X., Canagaratna, M., Nowak, J., Daube, C., Yan, C., Nie, W., Onasch, T., Jayne, J., Kolb, C., Davidovits, P., Worsnop, D. and Brune, W.: Controlled nitric oxide production via O(1D) + N2O reactions for use in oxidation flow reactor studies, Atmospheric Meas. Tech., 10(6), 2283–2298, doi:10.5194/amt-10-2283-2017, 2017.

Lambe, A. T., Ahern, A. T., Williams, L. R., Slowik, J. G., Wong, J. P. S., Abbatt, J. P. D., Brune, W. H., Ng, N. L., Wright, J. P., Croasdale, D. R., Worsnop, D. R., Davidovits, P. and Onasch, T. B.: Characterization of aerosol photooxidation flow reactors: heterogeneous oxidation, secondary organic aerosol formation and cloud condensation nuclei activity measurements, Atmospheric Meas. Tech., 4(3), 445–461, doi:10.5194/amt-4-445-2011, 2011.

Li, R., Palm, B. B., Ortega, A. M., Hlywiak, J., Hu, W., Peng, Z., Day, D. A., Knote, C., Brune, W. H., de Gouw, J. A. and Jimenez, J. L.: Modeling the Radical Chemistry in an Oxidation Flow Reactor: Radical Formation and Recycling, Sensitivities, and the OH Exposure Estimation Equation, J. Phys. Chem. A, 119(19), 4418–4432, doi:10.1021/jp509534k, 2015.

Palm, B. B., Campuzano-Jost, P., Day, D. A., Ortega, A. M., Fry, J. L., Brown, S. S., Zarzana, K. J., Dube, W., Wagner, N. L., Draper, D. C., Kaser, L., Jud, W., Karl, T., Hansel, A., Gutiérrez-Montes, C. and Jimenez, J. L.: Secondary organic aerosol formation from in situ OH, O3, and NO3 oxidation of ambient forest air in an oxidation flow reactor, Atmospheric Chem. Phys., 17(8), 5331–5354, doi:10.5194/acp-17-5331-2017, 2017.

Palm, B. B., de Sá, S. S., Day, D. A., Campuzano-Jost, P., Hu, W., Seco, R., Sjostedt, S. J., Park, J.-H., Guenther, A. B., Kim, S., Brito, J., Wurm, F., Artaxo, P., Thalman, R., Wang, J., Yee, L. D., Wernis, R., Isaacman-VanWertz, G., Goldstein, A. H., Liu, Y., Springston, S. R., Souza, R., Newburn, M. K., Alexander, M. L., Martin, S. T. and Jimenez, J. L.: Secondary organic aerosol formation from ambient air in an oxidation flow reactor in central Amazonia, Atmospheric Chem. Phys., 18(1), 467–493, doi:10.5194/acp-18-467-2018, 2018.

Peng, Z. and Jimenez, J. L.: Modeling of the chemistry in oxidation flow reactors with high initial NO, Atmospheric Chem. Phys., 17(19), 11991–12010, doi:10.5194/acp-17-11991-2017, 2017.

Peng, Z., Day, D. A., Stark, H., Li, R., Lee-Taylor, J., Palm, B. B., Brune, W. H. and Jimenez, J. L.: HOx radical chemistry in oxidation flow reactors with low-pressure mercury lamps

systematically examined by modeling, Atmospheric Meas. Tech., 8(11), 4863–4890, doi:10.5194/amt-8-4863-2015, 2015.

Peng, Z., Day, D. A., Ortega, A. M., Palm, B. B., Hu, W., Stark, H., Li, R., Tsigaridis, K., Brune, W. H. and Jimenez, J. L.: Non-OH chemistry in oxidation flow reactors for the study of atmospheric chemistry systematically examined by modeling, Atmospheric Chem. Phys., 16(7), 4283–4305, doi:10.5194/acp-16-4283-2016, 2016.

Peng, Z., Palm, B. B., Day, D. A., Talukdar, R. K., Hu, W., Lambe, A. T., Brune, W. H. and Jimenez, J. L.: Model Evaluation of New Techniques for Maintaining High-NO Conditions in Oxidation Flow Reactors for the Study of OH-Initiated Atmospheric Chemistry, ACS Earth Space Chem., doi:10.1021/acsearthspacechem.7b00070, 2017.

---

## Author Comment (AC2) · 22 Feb 2018

**RESPONSE TO REVIEWER 2: Manuscript ID: amt-2017-352**

*Mitroo et al. introduce a method to deconvolve the measured residence time distribution (RTD) from sampling tubes to get the real RTD inside the Washington University Potential Aerosol Mass (WU-PAM) reactor, which is also validated by computational fluid dynamic (CFD) simulation. The idea of this paper can help improve the understanding of RTD for the oxidative flow reactor (OFR) user community. This paper is well-written and fits the scope of AMT. I suggest for publication after considering the following aspects:*

> We thank the referee for their time in reading and reviewing this manuscript.

**[In reference to: General Comments]**

*1. I agree with Review #2's comments about the expansion of Section "5 Potential implication". My concern is that how your method can be applied to simulation rather than just used to explain RTD. In other words, how does the incorporation of CSTR tank-inseries (TIS) model framework behave when compared with the PFR framework? For example, most of your inversion results indicate that the number of TIS, N, is a little bit larger than 1. Does that mean it is CSTR rather than PFR that can better represent OFR? So to simulate what happens in OFR, we should use CSTR model instead of PFR? Then the question is to what extent the difference will be introduced to the simulated results by shifting from PFR to CSTR. I think the authors should clarify these points in this section.*

> The referee is correct that given the number of TIS is a little bit larger than 1, the reactor is considered to behave more like a CSTR than as a PFR. We address comments for both referees in the revised manuscript (lines 406-434):
>
> "Initial PAM modeling work assumed plug flow behavior in OFRs (Li et al., 2015). Li et al. stated that correcting for the non-ideal E-Curve in their OFR would account for ~10% error in their oxidant exposure results, which is less than the overall model uncertainty. However, recent work incorporates the effect of non-ideal RTDs on model outputs (Palm et al., 2017, 2018, Peng et al., 2015, 2016; Peng and Jimenez, 2017). Peng et al. (2015) show that for three OFR operational modes (that is, modes of different oxidant formation mechanisms denoted by 'OFR185', 'OFR254-70', and 'OFR254-7'), a comparison between model output for ideal plug flow vs. non-ideal RTDs (using the RTD experimentally obtained by Lambe et al., 2011a) for OH exposure ($OH_{exp}$) generally agree within a factor of 2 for low $OH_{exp}$; the model disagreement exacerbates at high $OH_{exp}$ beyond a factor of ~4. Peng and Jimenez then extend OFR operational modes to include N-containing chemistry (in modes referred to therein as 'OFR185-iNO', 'OFR185-7-iNO', and 'OFR185-70-iNO') where at moderate-to-high $OH_{exp}$, the deviations exacerbate significantly, although the authors argue those conditions represent unrealistic chemical pathways. It is worthwhile noting that the chemistry modeled by Peng and Jimenez may find a workaround by utilizing $N_2O$ as NO precursor (Lambe et al., 2017) rather than NO itself, potentially minimizing RTD-related errors. Palm et al. (2018) report data from OFR field deployment where the same comparison (ideal plug flow vs. the RTD experimentally obtained by Lambe et al., 2011a) suggests RTD-related errors overpredict (for CO) or

underpredict (for toluene and monoterpenes) photochemical age (that is, the ratio of OHexp to tropospheric average OH number concentrations) in the reactor, generally within a factor of 3 of model error. Considering this work employs the compartmental model RTD described by Lambe et al. (2011a), which for reasons mentioned in the previous section may not be the true PAM RTD, and given that non-ideality in RTDs affects certain OFRs more than others, implementing the method presented here to obtain a more representative reactor RTD can either help constrain error uncertainty in the models, or possibly extend the $OH_{exp}$ range in which OFRs can be operated, a reportedly nontrivial task (Palm et al., 2018). Considering our results indicate that OFRs like the WU-PAM exhibit an RTD closely matching that of an ideal CSTR, which is more well-mixed than the Lambe et al. RTD, the sensitivity analysis conducted so far could represent a lower bound for error analysis because the Lambe et al. RTD is closer to a PFR-like RTD than a CSTR-like RTD."

*2. TIS model can have different forms. The authors assume the same residence time for each CSTR-tank and find the tank number. One can also take the form with a fixed CSTR (or PFR or mixed CSTR/PFR) number but to find each residence time, which looks more reasonable given the CFD simulation. Can the authors discuss this a little bit more? For example: How does the number of TIS, N, depend on the flow rate, or in other words the average residence time? Since the flow rate changes the fluid field, the mixing style could be different at different flow rates (e.g. Fig.3a-c). But I cannot see any trend. Can the author give some explanation for that?.*

The classic TIS model assumes constant mean residence time across the $N$ tanks, which is $\bar{t}_i = \bar{t}/N$. In this work it is treated as a two-parameter model in which both $N$ and $\bar{t}$ are scanned to find the optimum value pair that results in the best fit with experimental data. It is found that the calculated mean residence time $\bar{t}$ is similar to the space-time $\tau$ as expected. Theoretically we can use any well-defined reactor model in place of TIS, such as the axial dispersion model (ADM) (employed by Lambe et al., 2011a in tandem with compartmental modeling) which measures the non-ideality from PFR. Mixed CSTR/PFR is also possible, provided the mathematical derivation is properly carried out. Developing such a new model is out of the scope of the current manuscript, but can be recommended as future work in this field. Whether the reactor model selected is valid to represent the real reactor is subject to validation with experimental measurements, as performed in Figure 3. In this work we find the TIS model is satisfactory for the PAM reactor according to the close agreement between model prediction and experimental data, stating the caveat that the TIS model is *not* phenomenological. The reviewer raised the question about unclear trend between $N$ and space-time in Figure 3(a-c), which is interesting to the authors too. Our guess is that under these conditions the reactor behaves so similarly to a single CSTR that the subtle differences are buried in the experimental uncertainties. Perhaps the trend would become more clear as the space-time is further raised, and we predict the trend to be $N$ increasing with space-time. The reason is that the larger the space-time, the slower the flow, thus the weaker the turbulence and back-mixing, which means the further away the reactor is from a single CSTR. This reasoning is backed by Figure 3f, where $N$ is more than doubled at a much higher space-time (although it is also a different configuration).

*3. Equations in Appendix B should be carefully checked. For example, in Eq. (B3) it should be $E_l/_{N-i}$ instead of $E_l/_{N-1}$. In addition, try to avoid "N", since the number of TIS is also "N", which may cause confusion. In Eq. (B8), **A** is a matrix, which should be listed as $A_{i,j}$ not just $A_{N,i}$. One more, as a vector, **B** should be listed as $B_i$, with i = 1, 2, ..., N - 1 and i = N. About time step $\Delta t$, see following comments for Figure 3.*

We thank the referee for pointing out the inconsistent notations in the equations. The misuse of indices can make the equations confusing and even wrong. We rewrote all of the equations in Appendix B with carefully checked syntax. We hope they bring much more clarity now. Please see revised manuscript lines 608-672.

**[In reference to: Specific Comments]**

*1. Line 182: "create allow", delete either one.*

Similar to Reviewer 1, we thank the referee for pointing this out, and have deleted "allow".

*2. Line 218: "F-curve", define it here or mention it later*

We restructured the sentences (L226-227 of the revised manuscript):

"After the simulation, the exit concentration is mixing-cup averaged to output a representative of a cumulative RTD (explained in the next section)."

*3. It is unnecessary to list both dimensional and non-dimensional equations at the same time, e.g. EQ. 1-4 and 8-9, since the non-dimensional form has been introduced in detail in Appendix A*

We respect the referee's point of view, however we choose to represent both dimensional and dimensionless equations in the main text for the audience.

*4. Figure 2: Please use higher resolution figures and rearrange the figure locations (too compact, and x-labels are hidden)*

We restructured Fig. 2 accordingly, incorporating requests from Reviewer 1 as well.

*5. Figure 3: Why time resolution is different in Panel e? Does $\Delta t$ in Appendix B correspond to the time interval in Figures 3a-f?*

We had instrument problems that day, and could not take datapoints as frequently as for all other panels. We have noted it in the figure caption. This doesn't affect the output of our algorithm based on longer $\Delta t$.

"Lower frequency data for panel e) was due to instrument repair, and temporarily set on longer averages."

*6. Figure 5: Please use an intuitive y-label instead of "F". Also please specify "N" value in the caption*

We relabeled the y-axis with "Normalized Concentration" for easier interpretation The "N" for the N-CSTR acronym the referee is referring to in the legend is another acronym for the TIS model. We clarifiy this in the caption, and introduced this acronym along 'TIS' in Section 3.2 (lines 313-317):

"We chose to apply the tank-in-series (TIS) model (MacMullin and Weber Jr., 1935), also referred to as N-CSTR model, to the convolution integral since it is a one parameter model that, although not specific to flowtube, tubular, laminar, or plug-flow reactors, gives an idea of where the reactor lies on the spectrum of mixed flow vs. plugged flow based on the value of a parameter, *N*"

**References**

Lambe, A., Massoli, P., Zhang, X., Canagaratna, M., Nowak, J., Daube, C., Yan, C., Nie, W., Onasch, T., Jayne, J., Kolb, C., Davidovits, P., Worsnop, D. and Brune, W.: Controlled nitric oxide production via O(1D) + N2O reactions for use in oxidation flow reactor studies, Atmospheric Meas. Tech., 10(6), 2283–2298, doi:10.5194/amt-10-2283-2017, 2017.

Li, R., Palm, B. B., Ortega, A. M., Hlywiak, J., Hu, W., Peng, Z., Day, D. A., Knote, C., Brune, W. H., de Gouw, J. A. and Jimenez, J. L.: Modeling the Radical Chemistry in an Oxidation Flow Reactor: Radical Formation and Recycling, Sensitivies, and the OH Exposure Estimation Equation, J. Phys. Chem. A, 119(19), 4418–4432, doi:10.1021/jp509534k, 2015.

Palm, B. B., Campuzano-Jost, P., Day, D. A., Ortega, A. M., Fry, J. L., Brown, S. S., Zarzana, K. J., Dube, W., Wagner, N. L., Draper, D. C., Kaser, L., Jud, W., Karl, T., Hansel, A., Gutiérrez-Montes, C. and Jimenez, J. L.: Secondary organic aerosol formation from in situ OH, O3, and NO3 oxidation of ambient forest air in an oxidation flow reactor, Atmospheric Chem. Phys., 17(8), 5331–5354, doi:10.5194/acp-17-5331-2017, 2017.

Palm, B. B., de Sá, S. S., Day, D. A., Campuzano-Jost, P., Hu, W., Seco, R., Sjostedt, S. J., Park, J.-H., Guenther, A. B., Kim, S., Brito, J., Wurm, F., Artaxo, P., Thalman, R., Wang, J., Yee, L. D., Wernis, R., Isaacman-VanWertz, G., Goldstein, A. H., Liu, Y., Springston, S. R., Souza, R., Newburn, M. K., Alexander, M. L., Martin, S. T. and Jimenez, J. L.: Secondary organic aerosol formation from ambient air in an oxidation flow reactor in central Amazonia, Atmospheric Chem. Phys., 18(1), 467–493, doi:10.5194/acp-18-467-2018, 2018.

Peng, Z. and Jimenez, J. L.: Modeling of the chemistry in oxidation flow reactors with high initial NO, Atmospheric Chem. Phys., 17(19), 11991–12010, doi:10.5194/acp-17-11991-2017, 2017.

Peng, Z., Day, D. A., Stark, H., Li, R., Lee-Taylor, J., Palm, B. B., Brune, W. H. and Jimenez, J. L.: HOx radical chemistry in oxidation flow reactors with low-pressure mercury lamps systematically examined by modeling, Atmospheric Meas. Tech., 8(11), 4863–4890, doi:10.5194/amt-8-4863-2015, 2015.

Peng, Z., Day, D. A., Ortega, A. M., Palm, B. B., Hu, W., Stark, H., Li, R., Tsigaridis, K., Brune, W. H. and Jimenez, J. L.: Non-OH chemistry in oxidation flow reactors for the study of atmospheric chemistry systematically examined by modeling, Atmospheric Chem. Phys., 16(7), 4283–4305, doi:10.5194/acp-16-4283-2016, 2016.

Peng, Z., Palm, B. B., Day, D. A., Talukdar, R. K., Hu, W., Lambe, A. T., Brune, W. H. and Jimenez, J. L.: Model Evaluation of New Techniques for Maintaining High-NO Conditions in Oxidation Flow Reactors for the Study of OH-Initiated Atmospheric Chemistry, ACS Earth Space Chem., doi:10.1021/acsearthspacechem.7b00070, 2017.